# On the Birth of the Universe and Time

Natalia Gorobey [1], Alexander Lukyanenko [1] and Alexander V. Goltsev [2],*

[1] Department of Physics, Physical-Mechanical Institute, Peter the Great Saint Petersburg Polytechnic University, Polytekhnicheskaya 29, 195251 Saint Petersburg, Russia
[2] Ioffe Physical-Technical Institute, Polytekhnicheskaya 26, 195251 Saint Petersburg, Russia
* Correspondence: goltsev@ua.pt

**Abstract:** A theory of the initial state of the universe is proposed within the framework of the Euclidean quantum theory of gravity. The theory is based on a quantum representation in which the action functional is implemented as an operator on the space of wave functionals depending on the 4$D$ space metric and matter fields. The initial construction object is the eigenvalue of the action operator in the area of the origin of the universe with the given values of the 3$D$ metric and matter fields on the boundary. The wave function of the initial state is plotted as an exponential of this eigenvalue, after a Wick rotation in the complex plane of the radial variable of the Euclidean 4$D$ space. An estimate of the initial radius of the universe is proposed.

**Keywords:** universe; Einstein-Hilbert action; De Donder-Weyl canonical representation; Euclidean quantum gravity; time

## 1. Introduction

The works [1,2] laid the foundation for the development of the idea of the quantum birth of the universe from "nothing". In the work of Hartl and Hawking [1], a special solution of the equations of the quantum theory of gravity (the Wheeler–De Witt equation (WDW) [3,4])

$$\widehat{H}\psi = \widehat{H}^i\psi = 0 \qquad (1)$$

is defined for the wave function of the universe $\psi$ in the form of a Euclidean functional integral over all Riemannian geometries (and fields of matter) with given boundary values on a (single) 3$D$ spatial section $\Sigma$ (no-boundary wave function). In [2], for a similar solution, a visual representation was proposed in the form of the amplitude of quantum tunneling from zero to a finite radius of the 3$D$ spatial section of the universe $\Sigma$. In both cases (tunnel and no-boundary), the wave function is calculated at the saddle point (instanton) of the Euclidean action. Within the framework of the semiclassical approximation, in the classically allowed region of the dynamics of the universe, one can determine the classical time parameter. The boundary of the classically allowed region of motion with real time is called the "bounce" point of the universe. For a homogeneous model of the universe with a cosmological constant and a scalar field of matter, the regularities of the formation of the inflationary stage of the expansion of the universe immediately after the "bounce" point were studied in [5,6]. The dependence of the position of this point on the initial value of the scalar field at zero radius (the "south pole" of the universe) is found. The results obtained in the semiclassical approximation for the Euclidean functional integral can also be found directly from the WDW equation without resorting to the functional integral [7]. This transformation of the approach is caused by the fact that the Euclidean quantum theory of gravity in terms of the functional integral [8] turns out to be untenable in the general case (outside the semiclassical approximation) due to the sign indefiniteness of the Euclidean action of the theory of gravity and, as a consequence, the divergence of the integral on the space of Riemannian 4$D$ metrics. Thus, time as a parameter of evolution is not defined in the modern quantum theory of gravity (QG). In [9], an alternative formulation of the

QG was proposed in terms of the wave functional defined on all pseudo-Euclidean $4D$ metrics (and matter fields) bounded by the initial and final spatial sections $\Sigma$. In the new formulation, the wave functional is defined by a secular equation for the action operator in the space of world histories of the universe. Note that the quantization of action together with energy is also considered in the alternative Hamilton–Jacobi formulation of stationary quantum mechanics, in which time can be introduced as an additional parameter (see [10]). In our case, the action operator has a different meaning, and the formulation itself is equivalent to quantum mechanics based on the Schrödinger equation (see [9]) together with all well-known consequences. In this case, the wave functional is equal (in a discrete approximation) to the product of the wave functions taken at all times. Being in QG an invariant of general covariant transformations of the world history of the universe, wave functional allows us to introduce the evolution parameter as the average geodesic distance between boundary spatial sections. In the new formulation of the QG dynamics, the question of the initial state of the universe, which must be determined on the initial spatial section $\Sigma_0$, remains open.

In this work, the quantum state of the universe on the initial spatial section $\Sigma_0$ (Beginning of the universe) is found using the secular equation for the action operator of the theory of gravity in Euclidean form. This operator is defined on the set of $4D$ Riemannian metrics with given boundary values for the metric and matter fields on $\Sigma_0$. Thus, a unified approach is proposed for formulating the dynamics and determining the Beginning of the universe based on the Hilbert–Einstein action of General Relativity. In both cases, to determine the action operator, the canonical form of the Hilbert–Einstein action is taken as the initial one. In this work, we will not consider f(R)-theory of gravity (see [9] on this topic). However, in the case of the Euclidean action, where all four coordinates of the $4D$ manifold are completely equal, a modification of its canonical structure, first formulated by De Donder and Weyl (DDW) [11,12], is required. The modification of the canonical quantization rules proposed in [9] turns out to be applicable to this modified canonical form of the original Hilbert–Einstein action. We would like to note that our work lies in the field of nonpertrubative quantum gravity theory (as applied to cosmology), where there is no renormalizability problem. A possible connection with (super)string theory is the subject of further research.

In the next section, the modified canonical form of the DDW and its quantization are considered for a real scalar field. In the second section, the modified canonical form of DDW is obtained for the Hilbert–Einstein action. On its basis, in the third section, the quantum principle of least action (QPLA) for the Euclidean QG is formulated and the initial state of the universe is determined.

## 2. De-Donder–Weyl Canonical Form of the Action of a Scalar Field

As the simplest example of the modified De Donder–Weyl (DDW) canonical form, consider it for the Euclidean action of a scalar field,

$$I_E[\varphi] = \int \sqrt{g}\, d^4 x \left[ \frac{1}{2} g^{ik} \partial_i \varphi \partial_k \varphi + V(\varphi) \right], \tag{2}$$

where $g_{ik}(x)$ is the Riemannian metric of the $4D$ manifold with signature $(+,+,+,+)$ and $g = \det g_{ik}$. There is no distinguished parameter of coordinate time here, and it would be natural to introduce generalized canonical momenta for all coordinates:

$$p^i(x) \equiv \frac{\delta I_E}{\delta \partial_i \varphi(x)} = \sqrt{g}\, g^{ik} \partial_k \varphi. \tag{3}$$

Using the generalized Legendre transform, we introduce the generalized Hamilton functional,

$$
\begin{aligned}
H\left[p^i, \varphi\right] &= \int d^4 x\, p^i \partial_i \varphi(x) - I_E[\varphi] \\
&= \int \sqrt{g}\, d^4 x\left[\frac{1}{2g} g_{ik} p^i p^k - V(\varphi)\right],
\end{aligned}
\tag{4}
$$

and write action Equation (2) in the modified canonical form:

$$
I_E\left[p^i, \varphi\right] = \int d^4 x\, p^i \partial_i \varphi(x) - H\left[p^i, \varphi\right].
\tag{5}
$$

It is easy to see that the extremum of action Equation (5) over all variables gives the original equations for the scalar field.

The quantization of the "dynamics" of a scalar field in this modified canonical form is possible using the modified canonical quantization rules formulated in [9]. We formulate them here, remaining in the Euclidean form of the modified canonical action of the scalar field Equation (5). The quantum state of the field $\varphi(x)$ is now described by the wave functional $\Psi[\varphi(x)]$. For the quantum realization of generalized canonical momenta Equation (3), we take into account that the usual canonical momentum $\pi$ in pseudo-Euclidean space-time, corresponding to the generalized coordinate $q$, is replaced by $-i\pi_E$ in the transition to the imaginary time $t = i\tau_E$, and the original action describing the dynamics in real time $t$, is replaced by $-iI_E$. This means that the operator canonical representation of momentum on the space of wave functions in the Euclidean form of the theory has the form:

$$
\widehat{\pi}_E = -\hbar \frac{\partial}{\partial q}.
\tag{6}
$$

In accordance with this, the generalized operator representation of the canonical momenta of the DDW Equation (3) on the space of wave functionals $\Psi[\varphi(x)]$ has the form [9]:

$$
\widehat{p}^i(x)\Psi = -\widetilde{\hbar}^i \frac{\delta\Psi}{\delta_i \varphi(x)},
\tag{7}
$$

where

$$
\widetilde{\hbar}^i = \hbar \epsilon^i,
\tag{8}
$$

and $\epsilon^i$ are constant length dimensions. We will reveal the meaning of these constant and variational derivatives in Equation (7) using the lattice approximation to describe the state of the field $\varphi(x)$. We fix coordinates in $4D$ space and introduce a set of points $\overrightarrow{x}^a$ forming a lattice with a unit cell in the form of a parallelepiped with edges $\overrightarrow{\epsilon}^i$ of length $\left|\overrightarrow{\epsilon}^i\right| = \epsilon^i$:

$$
\overrightarrow{x}^a = \sum_{i=1}^{4} n_i^a \overrightarrow{\epsilon}^i,
\tag{9}
$$

where $a$ is the lattice node number given by a set of integers $a \equiv \left\{n_i^a\right\}$. Let us replace (approximate) the continuous field $\varphi(x)$ by the set of its values $\varphi_a$ at each vertex of the lattice $\overrightarrow{x}^a$. Let us also approximate the wave functional $\Psi[\varphi(x)]$ by a function of several variables $\Psi(\varphi_a)$—the values of the field at the lattice vertices $\varphi_a$. Taking into account the connection between the variational derivative of the functional and the partial derivative of its lattice approximation in the case of a function of one variable $\varphi(t)$ [13],

$$
\frac{\delta\Psi}{\delta\varphi(t_a)} = \frac{1}{\varepsilon} \frac{\partial\Psi}{\partial\varphi_a},
\tag{10}
$$

the partial variational derivative of the wave functional in Equation (7), for example, in the direction $i$, is defined as follows (assuming also the lattice approximation of the partial derivative):

$$\frac{\delta \Psi}{\delta_i \varphi(\overrightarrow{x}^a)} = \frac{1}{\epsilon_i} \frac{\Psi\big(\varphi_{c \neq a}, \varphi\big(\overrightarrow{x}^a + \overrightarrow{\epsilon}^i\big)\big) - \Psi\big(\varphi_{c \neq a}, \varphi_a\big)}{\varphi\big(\overrightarrow{x}^a + \overrightarrow{\epsilon}^i\big) - \varphi_a}. \tag{11}$$

Then the lattice realization of the generalized canonical momentum Equation (3) will be the fraction:

$$\widehat{p}^i(\overrightarrow{x}_a)\Psi = -\hbar \frac{\Psi\big(\varphi_{c \neq a}, \varphi\big(\overrightarrow{x}^a + \overrightarrow{\epsilon}^i\big)\big) - \Psi\big(\varphi_{c \neq a}, \varphi_a\big)}{\varphi\big(\overrightarrow{x}^a + \overrightarrow{\epsilon}^i\big) - \varphi_a}. \tag{12}$$

We also write in the lattice approximation:

$$\partial_k \varphi(\overrightarrow{x}_a) = \frac{\varphi\big(\overrightarrow{x}^a + \overrightarrow{\epsilon}^i\big) - \varphi_a}{\epsilon^k}. \tag{13}$$

Finally, approximating the integral in Equation (5) by the integral sum over the lattice, we introduce the operator of action on the lattice corresponding to Equation (5):

$$
\begin{aligned}
\widehat{I}_E \Psi &= -\hbar \sum_a \prod_i \epsilon^i \sum_k \frac{1}{\epsilon^k} \Big[ \Psi\Big(\varphi_{c \neq a}, \varphi\big(\overrightarrow{x}^a + \overrightarrow{\epsilon}^k\big)\Big) \\
&\quad - \Psi\big(\varphi_{c \neq a}, \varphi_a\big)\Big] - H\big(\widehat{p}^i, \varphi\big)\Psi.
\end{aligned}
\tag{14}
$$

It is understood that the lattice approximation in all these definitions becomes more accurate as $\epsilon^i \longrightarrow 0$.

The action operator Equation (14), according to [9], allows us to formulate the Euclidean quantum "dynamics" of a scalar field in the form of the corresponding secular equation:

$$\widehat{I}_E \Psi = \Lambda_E \Psi. \tag{15}$$

In [9] this formulation of dynamics is called the quantum principle of least action (QPLA). In what follows, we will be interested in the eigenvalue $\Lambda_E$ of the action operator. We will discuss its meaning after the formulation of a similar structure for the Riemannian metric field $g$ and the full formulation of the QPLA for Euclidean quantum gravity.

## 3. Canonical De Donder–Weyl Form of the Hilbert–Einstein Action

We will base the Euclidean QG on the modified canonical form of the DDW of the Hilbert–Einstein action, since it reflects the fact that there is no distinguished coordinate that can be associated with coordinate time. In this case, we are not confused by the violation of the general covariance, which will manifest itself in the appearance of additional coordinate conditions. We start the construction from the Euclidean form of the Hilbert–Einstein action (at this stage we do not take into account the matter field) [8],

$$I_{gE} = \frac{1}{4\pi} \int \sqrt{g} R d^4 x, \tag{16}$$

(we set $c = G = 1$). Lagrangian density

$$
\begin{aligned}
\pounds_g &= \frac{1}{4\pi}\sqrt{g}R = \frac{1}{4\pi}\sqrt{g}g^{ik}\Big(\partial_l \Gamma^l_{ik} - \partial_i \Gamma^l_{kl} \\
&\quad + \Gamma^l_{ik}\Gamma^m_{lm} - \Gamma^l_{im}\Gamma^m_{kl}\Big),
\end{aligned}
\tag{17}
$$

where $\Gamma_{ik}^l$ are Christoffel symbols [14]. We write it as follows:

$$
\begin{aligned}
\pounds_g \;=\; & \frac{1}{4\pi}\Big\{ \partial_l \Big[ \sqrt{g}\Big( g^{ik}\Gamma_{ik}^l - g^{il}\Gamma_{im}^m \Big) \Big] \\
& + \sqrt{g}\Big[ \Big( \frac{1}{2}g^{\alpha\beta}g^{i\gamma} - g^{\alpha i}g^{\beta\gamma} \Big)\Gamma_{il}^l \\
& + \Big( \frac{1}{2}g^{\alpha\beta}g^{ik} - g^{\alpha i}g^{\beta k} \Big)\Gamma_{ik}^\gamma \Big]\partial_\gamma g_{\alpha\beta} \\
& + \sqrt{g}\,g^{ik}\Big( \Gamma_{ik}^l\Gamma_{lm}^m - \Gamma_{im}^l\Gamma_{kl}^m \Big) \Big\}.
\end{aligned}
\tag{18}
$$

We define the generalized momenta $P^{\gamma|\alpha\beta}$ conjugate to the components of the Riemannian metric $g_{\alpha\beta}$ as partial derivatives of the Lagrange density of action Equation (16) with respect to $\partial_\gamma g_{\alpha\beta}$. The total divergence (the first term in Equation (18)) is not affected in this case, as well as the Christoffel symbols, because the derivatives with respect to $\Gamma_{ik}^l$ are reduced to the identities that define them. As a result, we obtain:

$$
\begin{aligned}
P^{\gamma|\alpha\beta} \;=\; & \frac{1}{4\pi}\sqrt{g}\Big[ \Big( \frac{1}{2}g^{\alpha\beta}g^{i\gamma} - g^{\alpha i}g^{\beta\gamma} \Big)\Gamma_{il}^l \\
& \Big( \frac{1}{2}g^{\alpha\beta}g^{i\gamma} - g^{\alpha i}g^{\beta\gamma} \Big)\Gamma_{il}^l \Big].
\end{aligned}
\tag{19}
$$

Expression Equation (19) is not a tensor, so our constructions are not covariant from the very beginning. As a consequence, we obtain additional conditions that explicitly violate covariance. It is easy to check that the generalized momenta Equation (19) obey the identities:

$$
P^{\gamma|\alpha\beta}\left( g_{\gamma\delta}g_{\alpha\beta} - 2g_{\alpha\gamma}g_{\beta\delta} \right) = 0.
\tag{20}
$$

They must be taken into account when trying to solve Equation (19) with respect to $\Gamma_{ik}^l$. We have:

$$
\begin{aligned}
& \frac{16\pi}{\sqrt{g}}P^{\gamma|\alpha\beta}\,g_{\gamma p}g_{\alpha q}g_{\beta r} \\
=\; & \Gamma_{p|qr} + \frac{1}{2}g_{qr}\Big( g^{km}\Gamma_{m|pk} - g^{ik}\Gamma_{p|ik} \Big) \\
& - g_{pr}g^{km}\Gamma_{m|qk}.
\end{aligned}
\tag{21}
$$

Then, taking into account Equation (21), identities Equation (20) lead to additional coordinate conditions

$$
\partial_i g = 0.
\tag{22}
$$

This means that the canonical equality of coordinates in the DDW representation can be achieved if the determinant of the metric tensor is constant throughout the space. Under these additional conditions, from Equation (21) we obtain:

$$
\begin{aligned}
\Gamma_{p|qr} \;=\; & \frac{16\pi}{\sqrt{g}}P^{\gamma|\alpha\beta}\Big[ \big( g_{\gamma p}g_{\alpha q}g_{\beta r} - g_{\alpha\gamma}g_{\beta p}g_{qr} \big) \\
& - \frac{1}{3}g_{pr}\big( g_{\alpha q}g_{\beta\gamma} \big) \Big].
\end{aligned}
\tag{23}
$$

We are now ready to define the generalized Hamilton functional in the DDW representation using the generalized Legendre transform:

$$
\begin{aligned}
H_{gE}[P,g] &= \int_{\Omega} d^4x \partial_{\gamma} g_{\alpha\beta} P^{\gamma|\alpha\beta} - I_{gE} \\
&= \int_{\partial\Omega} dS_{\gamma} g_{\alpha\beta} P^{\gamma|\alpha\beta} \\
&\quad + \frac{1}{16\pi} \int_{\Omega} \sqrt{g} d^4x g^{ik} \left( \Gamma^l_{ik} \Gamma^m_{lm} - \Gamma^l_{im} \Gamma^m_{kl} \right).
\end{aligned}
\tag{24}
$$

Taking into account Equation (23), the second term in the Hamilton functional Equation (24) is the quadratic form of momenta:

$$
16\pi \int_{\Omega} \frac{d^4x}{\sqrt{g}} \left( g_{\alpha\alpha'} g_{\beta\beta'} g_{\gamma\gamma'} - g_{\alpha\gamma} g_{\alpha'\gamma'} g_{\beta\beta'} \right) P^{\gamma|\alpha\beta} P^{\gamma'|\alpha'\beta'}.
\tag{25}
$$

After adding the matter fields to the Riemannian metric $g$, the set of which we denote by the collective symbol $\varphi$, we can write down the generalized canonical form of the action of the Euclidean theory of gravity in the domain $\Omega$. We also take into account additional conditions Equation (20) on the generalized canonical variables of the metric field with the help of the corresponding Lagrange multipliers $\eta^{\delta}$. It is easy to see that conditions Equation (22) are satisfied automatically. Finally, the action of the DDW of the Euclidean theory of gravity takes the form:

$$
\begin{aligned}
I_E[P, g, p, \varphi, \eta] &= \int_{\Omega} d^4x \partial_{\gamma} g_{\alpha\beta} P^{\gamma|\alpha\beta} + \int_{\Omega} d^4x p^i \partial_i \varphi(x) \\
&\quad - H_{gE}[P,g] - H_{\varphi E}[g, p, \varphi] \\
&\quad \int_{\Omega} d^4x \eta^{\delta} \left( g_{\gamma\delta} g_{\alpha\beta} - 2g_{\alpha\gamma} g_{\beta\delta} \right) P^{\gamma|\alpha\beta}.
\end{aligned}
\tag{26}
$$

We will use this form of operation of the Euclidean theory of gravity as the basis for the QPLA for determining the initial quantum state of the universe in the next section.

## 4. The Beginning of the Universe and Time

Let us concretize the form of the Euclidean birth region of the universe $\Omega$. The coordinate conditions Equation (22) are also satisfied in the simplest case of a homogeneous isotropic Riemannian space, the Euclidean space. Let us introduce in this space the spherical coordinates $x_{\alpha} = (r, \theta_A)$, $A = 1, 2, 3$. The beginning of the radial coordinate $r = 0$ will be called the "south pole" of the universe in accordance with the terminology of [5,6]. Let $\Omega$ be a convex region covering the "south pole" and bounded by the surface $\rho = \rho(\theta)$. The radial variable $0 < r \leq \rho(\theta)$ will be singled out as one of the spatial coordinates near the boundary $\partial\Omega$ of the region, which we will further "join" with the time parameter of the universe. All Riemannian metrics $g_{\alpha\beta}$ satisfying conditions Equation (22) in the domain $\Omega$ will be considered related (by a determinant-preserving bijection) to the spherical coordinates of the Euclidean space. We introduce in these coordinates a spatial lattice $\overrightarrow{x}_a$ with constants $\epsilon^i$ (see Equation (9)).

The next step is quantization. At this step, we define the quantum version of the action functional Equation (26) in the form of a difference operator $I_E\left[\widehat{P}_a, g_a, \widehat{p}_a, \varphi_a, \eta_a\right]$ on the lattice $\overrightarrow{x}_a$, acting in the space of wave functionals (functions on the lattice) $\Psi(g_a, \varphi_a, \eta_a)$ (here we agree to place the variational differentiation operators on the right), and let us formulate the "dynamic" principle of the Euclidean quantum theory of gravity on the domain $\Omega$ in the form of a secular equation for the action operator:

$$
I_E\left[\widehat{P}_a, g_a, \widehat{p}_a, \varphi_a, \eta_a\right] \Psi = \Lambda_E \Psi.
\tag{27}
$$

This is a difference (matrix) equation for $\Psi(g_a, \varphi_a, \eta_a)$ with given field values $(g_a, \varphi_a, \eta_a)_{\partial\Omega}$ on the boundary $\partial\Omega$. Note that the first term on the right side of Equation (24) in the form of a surface integral over this boundary, when quantized, turns into a sum over lattice points satisfying the equation $r_a = \rho(\theta_a)$, partial derivatives (finite differences) of the wave functional $\Psi(g_a, \varphi_a, \eta_a)$ in the radial direction (along the normal to the boundary $\partial\Omega$). According to the QPND formulation [9], the eigenvalue $\Lambda_E$ of the action operator depends only on these boundary values. We will use this eigenvalue as the basis for determining the initial state of the universe on the boundary $\partial\Omega$, and further fix the boundary itself by an additional extremum principle. To do this, we recall [9] that for nonrelativistic quantum mechanics of a particle whose wave function is represented in exponential form

$$\psi(q, t) = \exp\left(\frac{i}{\hbar} R(q, t)\right),$$ (28)

in the formulation of the QPND on the interval of (real) time $[0, T]$, the eigenvalue of the action operator is equal to

$$\Lambda = R(q_T, T) - R(q_0, 0).$$ (29)

In the problem considered here, the action is Euclidean (imaginary time), and the initial value of the wave function with phase $R(q_0, 0)$ is absent by definition. From this it follows that the wave function of the Beginning of the Universe should be sought in the form of an exponential expression of the form:

$$\Phi_0(g_a, \varphi_a, \eta_a)_{\partial\Omega} = \exp\left[\frac{i}{\hbar} \Lambda_E(g_a, \varphi_a, \eta_a)_{\partial\Omega}\right].$$ (30)

This is all we need from the spectral problem Equation (27). The eigenwave functional $\Psi(g_a, \varphi_a, \eta_a)$ has the meaning of the probability amplitude of various Euclidean 4$D$ geometries (and matter fields) inside $\Omega$ having given values on the boundary $(g_a, \varphi_a, \eta_a)_{\partial\Omega}$. However, we do not need these probabilities at this stage. It is only important that all values are finite. It can be expected that this is so for sufficiently small values of $\rho$, where, as we know, solutions of the classical Euclidean Einstein equations (instantons) exist. With an increase in $\rho$, we reach the point of "return" of the classical Euclidean solutions and exit from the "tunnel" [2]. To find the spatial form of the instanton, i.e., the function $\rho(\theta_a)$, we formally "go out" into a domain with real time on the boundary $\Omega$. As real time, we will consider the continuation of the radial variable obtained by its Wick rotation in the complex plane $r$. Formally, this is achieved by the $3 + 1$ splitting of the 4$D$ metric of Arnowitt, Deser, and Mizner [15],

$$ds^2 = (Ndr)^2 + g_{AB}\left(d\theta^A + N^A dr\right)\left(d\theta^B + N^B dr\right),$$ (31)

and subsequent replacement $N \longrightarrow iN$ [9]. It suffices to do this on the boundary $\partial\Omega$, i.e., directly in the eigenvalue $\Lambda_E$. Inside the region $\Omega$, this rotation occurs automatically due to the constancy of the determinant $g = N^2 \det g_{AB}$. Taking into account that the action in the pseudo-Euclidean space $I_L$ with real time $t$ related to the Euclidean one as follows $t = i\tau_E$ after the Wick rotation is related to the Euclidean action $I_E$ by the relation $I_L = -iI_E$, and, accordingly, $\Lambda_L = -i\Lambda_E$, we obtain that the imaginary part of the exponent in Equation (30) (the phase of the wave functions of the universe) is proportional to

$$F = -\Lambda_E(iN, g_{AB}, \varphi, \eta)_{\partial\Omega}.$$ (32)

The phase of the wave function in quantum mechanics is the quantum analogue of the classical action, according to Dirac [16]. To determine the shape of the instanton corre-

sponding to the given boundary conditions $(g_a, \varphi_a, \eta_a)_{\partial\Omega}$, we will look for the minimum (extremum) of the functional Equation (32) with respect to the function $\rho(\theta_{Aa})$:

$$\frac{\delta F}{\delta \rho(\theta_{Aa})} = 0. \tag{33}$$

Let us recall in conclusion that the action Equation (26) and the action operator depend on the (real) indefinite Lagrange multipliers $\eta^{\delta}$, which take into account the constraints Equation (20) that have arisen in the DDW formalism in the theory of gravitation. We also fix them by additional extremum conditions

$$\frac{\delta F}{\delta \eta_a^{\delta}} = 0. \tag{34}$$

The solution of the system of Equations (33) and (34) should be substituted into Equation (30), as a result of which we obtain the wave function the Beginning of the universe

$$\psi_0(g_a, \varphi_a,)_{\partial\Omega} = \exp\left[\frac{i}{\hbar}\Lambda_E(g_a, \varphi_a)_{\partial\Omega}\right]. \tag{35}$$

## 5. Conclusions

The initial state of the universe obtained in this paper using the modified canonical form of the DDW in QPLA is the missing element for the complete formulation of the dynamics of QG in terms of the wave functional and the real time coordinate parameter [9]. Thus, in QG, the principle of general covariance is restored in its original sense as the independence of the laws of quantum dynamics from an arbitrary choice of space-time coordinates. In contrast, the QG, based on WDW, generally excludes the use of any external coordinate parameter of time and requires its (time) identification with one of the fundamental dynamical variables of the theory, which inevitably destroys covariance. The noncovariance of the initial state, Equation (35), associated with the additional condition Equation (22) should not bother us, since the observer cannot exist in the Euclidean region. As a solution to the system of Equations (33) and (34), the function $\rho(\theta_{Aa})$ is expressed in terms of the boundary values of the fundamental dynamic variables $(g_a, \varphi_a,)_{\partial\Omega}$. Nevertheless, we can say that the initial $3D$ hypersurface $\Sigma_0$, to which we refer these quantities, is located at a distance

$$r_0 = \frac{\left\langle \psi_0 \middle| \frac{1}{2\pi^2} \int_{\partial\Omega} \sqrt{\det g_{AB}} \rho(\theta_{Aa}) d^3\theta \middle| \psi_0 \right\rangle}{\langle \psi_0 | \psi_0 \rangle} \tag{36}$$

from the "south pole" of the universe in Euclidean space. Note that in average operation, integration is carried out also over the constant determinant $g$. This value can also be called the initial radius of the universe.

We emphasize once again that Formulas (28) and (29) explain how the final wave function is determined by the eigenvalue of the action operator in ordinary quantum mechanics. Based on this, we further construct the wave function of the Beginning of the Universe, see Equation (35), using the eigenvalue of the Euclidean action operator of the theory of gravity. In order to introduce time, we further need a Wick rotation of the radial variable of spherical coordinates at the "south pole" (the radial variable becomes the time parameter), as well as fixing the shape of the "initial instanton" by an additional extremum principle. Here we rely on the fact that the real part of the phase of the wave function in ordinary quantum mechanics is associated with the classical action (see [17]). Of course, these constructions must show their efficiency in specific approximations. The subsequent history of the universe with real time should be described, for example, using its wave functional, as proposed in [9]. At the present time we are looking for simple approximations to perform specific calculations.

The definition of the initial state completes the formulation of the covariant quantum dynamics of the universe in terms of the wave functional using arbitrary space-time coordinates [9]. Gravitational constraints—WDW equations, Equation (1)—are not used in this formalism. Note that the selection of time as a dynamic parameter means a violation of the general covariance, but in fact the general covariance in the proposed approach is violated: the use of the canonical De Donder–Weyl formalism in the case of the Hilbert–Einstein action leads, among other things, to an additional condition (22), which is not covariant. We consider the complete equality of the four coordinates at the "south pole" to be a more important requirement than the general covariance at the Beginning. However, the new approach to the formulation of the quantum dynamics of the universe does not preclude its description in terms of an internal parameter (multipoint, see [14]) of time. In this case, the question of identifying internal time remains open. One of the options for describing the dynamics of the universe in terms of internal time was proposed in [17], where one of the quantum numbers that arise when using the operator form of gravitational constraints is considered as internal time.

**Author Contributions:** Conceptualization, A.L.; writing—original draft preparation, N.G., A.L. and A.V.G. All authors have read and agreed to the published version of the manuscript.

**Funding:** This research received no external funding.

**Acknowledgments:** We would like to thank V. A. Franke for useful discussions.

**Conflicts of Interest:** The authors declare no conflict of interest.

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
