# Peer review of "On the Birth of the Universe and Time"

_universe, doi:10.3390/universe8110568_

Round 1

Reviewer 1 Report

Using the analogy with non relativistic quantum mechanics the authors reduce the problem of the initial state of the universe to an eigenvalue problem, where the boundary is fixed by an extremum principle. Then, the authors propose an exponential form for the wave function of the beginning of the universe.

The approach to the "beginning of time" is not clear, since it is codified within the restrictions (33) and (34). In order to explain the origin of time as a fundamental magnitude, it is necessary to sacrifice, in principle, the general covariance. 

However, the proposed approach enrich the discussion and may represent an advance in the problem of understanding the beginning of the universe.

Author Response

Our reply to remarks of Reviewer 1.

We appreciate Reviewer 1 for his/her positive estimation of our work and useful remarks. We have improved our manuscript by following these remarks.

Point 1. Reviewer 1 wrote:

“The approach to the "beginning of time" is not clear, since it is codified within the restrictions (33) and (34). In order to explain the origin of time as a fundamental magnitude, it is necessary to sacrifice, in principle, the general covariance.”

Response 1. Our reply:

In order to answer this remark, in the Conclusion we add a note:

“Note that the selection of time as a dynamic parameter means a violation of the general covariance, but in fact the general covariance in the proposed approach is violated: the use of the canonical DeDonder-Weyl formalism in the case of the Hilbert-Einstein action leads, among other things, to an additional condition, Eq. (22), which is not covariant. We consider the complete equality of the four coordinates at the “south pole” to be a more important requirement than the general covariance at the Beginning.”

Reviewer 2 Report

I found this article quite interesting.

The authors use their QM declared in previous publication( Universe 2021, 7(11), 452) for description of universe. In reality, we might be able defining
quantum state of a system for calculating mean values of the
observables over this state to compare them with the
observations. In previous article (Universe 2021, 7(11), 452)  authors
 have compared their quantization method with the canonical quantization
of a single particle.

1. In the submitted paper, the introductory section, in
which an ability of the author's approach to define quantum states
(in particular, vacuum state) of the quantized fields should be clarified.
It would be logical, if authors begin from the consideration of a single quantum
oscillator and define a vacuum state in their approach.

2. It should be noted, that their wave functional does not define quantum state
of a system and the authors have to describe a procedure how to
define a quantum state of a system by this functional.

3. A general formula for calculation of the mean values should be included.

4. At the end part of the article, mini-superspace example of  defining the initial state of universe looks logical. Besides, more references are needed (e.g. see Found. Phys. 47 (2017) 392-429).

The article could be published in the journal after revision proposed.

Author Response

Our reply to remarks of Reviewer 2.

We are grateful to Reviewer 2 for his/her positive estimation of our work and useful remarks. We have improved our manuscript by following these remarks.

Point 1. Reviewer 2 wrote:

“1. In the submitted paper, the introductory section, in
which an ability of the author's approach to define quantum states
(in particular, vacuum state) of the quantized fields should be clarified.
It would be logical, if authors begin from the consideration of a single quantum oscillator and define a vacuum state in their approach”.

Response 1. Our reply:

We added the following note to the introduction explaining the equivalence of the new formulation based on the action operator to ordinary quantum mechanics based on the Schrödinger equation. This also applies to all consequences, including for the harmonic oscillator:

“Note that the quantization of action together with energy is also considered in the alternative Hamilton-Jacobi formulation of stationary quantum mechanics, in which time can be introduced as an additional parameter (see Ref.[10]). In our case, the action operator has a different meaning, and the formulation itself is equivalent to quantum mechanics based on the Schrödinger equation (see Ref.[9]) together with all well-known consequences. In this case, the wave functional is equal (in a discrete approximation) to the product of the wave functions taken at all times”.

Point 2. Reviewer 2 wrote:

“2. It should be noted, that their wave functional does not define quantum state
of a system and the authors have to describe a procedure how to
define a quantum state of a system by this functional”.

Response 2. Our reply:

We have explained this above in our response to remark 1 (see also a corresponding note in the Introduction): the wave functional describes the quantum dynamics of the system. In the case of ordinary quantum mechanics, it reduces to the product of wave functions at each moment of time (see [9]).

Point 3. Reviewer 2 wrote:

“3. A general formula for calculation of the mean values should be included”.

Response 3.  Our reply:

The probability distribution on the configuration space of the system, determined by the wave functional, was introduced in [9]. In this paper, the average value of the instanton radius r_0 in the state ψ_0 is determined in the usual way, as described in any textbook of quantum mechanics. We give this formula in the text, see Eq. (36)

Point 4.

“4. At the end part of the article, mini-superspace example of defining the initial state of universe looks logical. Besides, more references are needed (e.g. see Found. Phys. 47 (2017) 392-429).”

Response 4. Our reply:

We agree that the next step should be to consider a simple model. Work in this direction is ongoing and requires some more time.

We are grateful to the referee for an interesting reference (Found. Phys. 47 (2017) 392-429) that we cite in the introduction:

“Note that the quantization of action together with energy is also considered in the alternative Hamilton-Jacobi formulation of stationary quantum mechanics, in which time can be introduced as an additional parameter (see [10]).”

Reviewer 3 Report

The paper is devoted  to the proposal of a theory of the initial state of the universe within the framework of the Euclidean quantum theory of gravity. It is based on a quantum representation in which the action functional is implemented as an operator on the space of wave functionals depending on the 4D space metric and matter fields.

The manuscript may be considered as the model description of derivation of the action functional (on the boundary of 4D Riemannian manifold) at the initial stage of Birth of Universe on a base of quantization of the action functional itself.  

 The authors in the manuscript appeal to the quantum theory of Einstein gravity.

 In this connection, one should  be noted that a  consistent theory of the quantum gravity which permits to get renormalized theory have not been yet constructed and, seemingly, can not be constructed outside the scope (super)string theory, especially on the initial stage of the Universe evolution. Therefore, a suggested model maybe by special low-energy approximation of real high-dimensional (d=10, 11 or 26) theory, in which among the matter fields one absent the other fields describing massless particles, e.g. photons and rest possible particles with higher helicities, as well as the rest massive particles with non-vanishing higher spins.

Besides, a modern research, which is applicable in addition to cosmology problems, includeswith various generalization of Einstein gravity in the form of  so called F(R)-gravity models.  Unfortunately, the manuscript does not touch this important aspects.

 By the next point is the use of the term “generalized canonical quantization”, based on the concept of authors from Ref. [9]. This term was reserved till 1977-1983 in the famous FIAN group papers by I.Batalin, E.Fradkin, G.Vilkovisski for canonical quantization of the dynamical systems subject to constraints.

Second, an idea of quantization based on the using as the Hamilton operator (related to the quantization the energy of the dynamical system)  the action functional itself acting on the space “quadratic integrable functionals” (instead of the wave functions) appears by original one but from my opinion requires detailed checking.

Derived final expression (35) for the wave functional of the Beginning of Universe  seems to be as undetermined. Explicit expression of  in (35) from my opinion has not concrete form, to make a conclusion what is made. It seems that the authors present the model with quantization the action as operator which acts in the space of “wave functionals” to avoid the extraction of a time component during working with Einstein gravity coupled with scalar matter  fields in terms of natural like system. All the calculations reduced to justification of the final expression for “wave function the Beginning of the universe in the form (35), without explicit definition of   .

The work obeys by the elements of formal consideration

I think the manuscript should be revised with taken into account the arguments above before the publication in the Universe.

Author Response

Our reply to remarks of Reviewer 3.

We are grateful to Reviewer 3 for his/her positive estimation of our work and useful remarks. We have improved our manuscript by following these remarks.

Point 1. Reviewer 3 wrote:

“ In this connection, one should  be noted that a  consistent theory of the quantum gravity which permits to get renormalized theory have not been yet constructed and, seemingly, can not be constructed outside the scope (super)string theory, especially on the initial stage of the Universe evolution. Therefore, a suggested model maybe by special low-energy approximation of real high-dimensional (d=10, 11 or 26) theory, in which among the matter fields one absent the other fields describing massless particles, e.g. photons and rest possible particles with higher helicities, as well as the rest massive particles with non-vanishing higher spins.”

Response 1. Our reply:

We would like to note that our work lies in the field of nonpertrubative quantum gravity theory (as applied to cosmology), where there is no renormalizability problem. A possible connection with (super)string theory may be a subject of further research. We have added the following note in the Introduction:

“We would like to note that our work lies in the field of nonpertrubative quantum gravity theory (as applied to cosmology), where there is no renormalizability problem. A possible connection with (super)string theory is the subject of further research.”

Reviewer 3 wrote:

“Besides, a modern research, which is applicable in addition to cosmology problems, includes with various generalization of Einstein gravity in the form of  so called F(R)-gravity models.  Unfortunately, the manuscript does not touch this important aspects.”

Response 1.  Our reply:

We thank Reviewer 3 for this comment.We inserted the following sentence in the introduction:

“In this work we will not consider f(R)-theory of gravity (see Ref. [9] on this topic)”.

In general, as stated in [9], one of the reasons for switching to a modified quantization procedure is the need to quantize time-nonlocal theories, which include F(R)-gravity models. Here we considered it inappropriate to complicate the consideration by such generalizations.

Point 2. Reviewer 3 wrote:

 “By the next point is the use of the term “generalized canonical quantization”, based on the concept of authors from Ref. [9]. This term was reserved till 1977-1983 in the famous FIAN group papers by I.Batalin, E.Fradkin, G.Vilkovisski for canonical quantization of the dynamical systems subject to constraints”.

Response 2. Our reply:

We are grateful to the referee for this remark. Throughout the text, “generalized canonical quantization” has been replaced by “modified canonical quantization”.

Point 2. Reviewer 3 wrote:

“Second, an idea of quantization based on the using as the Hamilton operator (related to the quantization the energy of the dynamical system)  the action functional itself acting on the space “quadratic integrable functionals” (instead of the wave functions) appears by original one but from my opinion requires detailed checking”.

Response 2. Our reply:

A detailed test of the approach that is developed in this article is that in ordinary quantum mechanics our approach is equivalent to describing quantum dynamics in terms of the Schrödinger equation for the wave function of the system. In this case, the wave functional is the product (the limit of the product) of the wave functions taken at each subsequent moment of time (see [9]). This clarification has been added to the introduction:

“Note that the quantization of action together with energy is also considered in the alternative Hamilton-Jacobi formulation of stationary quantum mechanics, in which time can be introduced as an additional parameter (see \cite{Floyd}). In our case, the action operator has a different meaning, and the formulation itself is equivalent to quantum mechanics based on the Schrödinger equation (see \cite{GLG}) together with all well-known consequences. In this case, the wave functional is equal (in a discrete approximation) to the product of the wave functions taken at all times.”

Point 3. Reviewer 3 wrote:

“Derived final expression (35) for the wave functional of the Beginning of Universe  seems to be as undetermined. Explicit expression of  in (35) from my opinion has not concrete form, to make a conclusion what is made. It seems that the authors present the model with quantization the action as operator which acts in the space of “wave functionals” to avoid the extraction of a time component during working with Einstein gravity coupled with scalar matter  fields in terms of natural like system. All the calculations reduced to justification of the final expression for “wave function the Beginning of the universe in the form (35), without explicit definition of”   .

Response 3. Our reply:

We have taken this remark into account by adding the following paragraph in the Conclusion:

“We emphasize once again that formulas Eqs. (28) and (29) explain how the final wave function is determined by the eigenvalue of the action operator in ordinary quantum mechanics. Based on this, we further construct the wave function of the Beginning of the Universe, Eq. (30), using the eigenvalue of the Euclidean action operator of the theory of gravity. In order to introduce time, we further need a Wick rotation of the radial variable of spherical coordinates at the "south pole" (the radial variable becomes the time parameter), as well as fixing the shape of the "initial instanton" by an additional extremum principle. Here we rely on the fact that the real part of the phase of the wave function in ordinary quantum mechanics is associated with the classical action (see Ref. [9]). Of course, these constructions must show their efficiency in specific approximations. The subsequent history of the universe with real time should be described, for example, using its wave functional, as proposed in Ref. [9]. At the present time we are looking for simple approximations to perform specific calculations”.

Point 4. Reviewer 3 wrote:

The work obeys by the elements of formal consideration.

Reponse 4. Our reply:

We agree with remark. We have noted this in the Conclusion:

“At the present time we are looking for simple approximations to perform specific calculations”.  

Round 2

Reviewer 2 Report

Revised version can be published as it is.

Reviewer 3 Report

After reading the author's remarks and corrections  I think that the manuscript can be accepted for publication in Universe